# The Interleukin-15 and Interleukin-8 Axis as a Novel Mechanism for Recurrent Chronic Rhinosinusitis with Nasal Polyps

**DOI:** 10.3390/biomedicines12050980

**Published:** 2024-04-29

**Authors:** Kai-Min Fang, Yen-Ling Chiu, Ruo-Wei Hong, Ping-Chia Cheng, Po-Wen Cheng, Li-Jen Liao

**Affiliations:** 1Department of Nursing, Oriental Institute of Technology, College of Healthcare and Management, New Taipei City 220, Taiwan; u701048@gmail.com; 2Department of Otolaryngology Head and Neck Surgery, Far Eastern Memorial Hospital, New Taipei City 220, Taiwan; i.cruising@gmail.com (P.-C.C.); powenjapan@yahoo.com.tw (P.-W.C.); 3Department of Internal Medicine, Far Eastern Memorial Hospital, New Taipei City 220, Taiwan; yenling.chiu@gmail.com (Y.-L.C.); rabit1989114@gmail.com (R.-W.H.); 4Department of Electrical Engineering, Yuan Ze University, Taoyuan 320, Taiwan

**Keywords:** chronic rhinosinusitis, cytokines, interleukin-15, recurrent nasal polyps

## Abstract

The prevention of postoperative recurrence after endoscopic sinus surgery (ESS) relies on targeting specific pathological mechanisms according to individuals’ immunological profiles. However, essential biomarkers and biological characteristics of difficult-to-treat chronic rhinosinusitis (CRS) patients are not well-defined. The aim of this study was to explore the immunologic profiles of subgroups of CRS patients and determine the specific cytokines responsible for recalcitrant or recurrent CRS with nasal polyposis (rCRSwNP). We used 30 cytokine antibody arrays to determine the key cytokines related to recurrent polypogenesis. Enzyme-linked immunosorbent assay (ELISA) experiments were conducted to assess the levels of these key cytokines in 78 patients. Polymorphonuclear leukocytes (PMNs) isolated from nasal polyps were challenged with specific cytokines to examine the levels of enhanced interleukin (IL)-8 production. Finally, we used immunohistochemistry (IHC) staining to check for the presence and distribution of the biomarkers within nasal polyps. A cytokine antibody array revealed that IL-8, IL-13, IL-15, and IL-20 were significantly higher in the recalcitrant CRSwNP group. Subsequent ELISA screening showed a stepwise increase in tissue IL-8 levels in the CHR, CRSsNP, and CRSwNP groups. PMNs isolated from nine CRSwNP cases all demonstrated enhanced IL-8 production after IL-15 treatment. IHC staining was labeled concurrent IL-8 and IL-15 expression in areas of prominent neutrophil infiltration. Our results suggest that IL-15 within the sinonasal mucosa plays a crucial role in promoting IL-8 secretion by infiltrating PMNs in recalcitrant nasal polyps. In addition, we propose a novel therapeutic strategy targeting the anti-IL-15/IL-8 axis to treat CRS with nasal polyposis.

## 1. Introduction

Chronic rhinosinusitis (CRS) is a highly heterogeneous and complex disease entity. Targeted multidirectional treatments do not always yield predictable outcomes, and endoscopic sinus surgery (ESS) remains the mainstay of treatment for patients with CRS, which is refractory to medical therapy. Difficult-to-treat CRS is still a challenge to rhinology surgeons. Despite aggressive surgical management, a considerable number of patients suffer from disease recurrence. In this subset of patients who are suffering from severe symptoms where the disease is uncontrolled by current guideline treatments, the further development of novel therapy is mandatory.

Nasal polyps (NPs) are frequently observed in CRS, and the occurrence of NPs indicates a more serious condition with a higher recurrence rate [1]. Accordingly, CRS is phenotyped as CRS with NPs (CRSwNP) and CRS without NPs (CRSsNP). Inflammatory patterns between these two diseases are distinct. In Western countries, CRSwNP appears to be dominated by T-helper cell (Th)2 responses and eosinophils, while CRSsNP is associated with Th1 inflammatory pathways [2]. However, subsequent studies revealed a wider spectrum of immunologic profiles in CRSwNP, especially in East Asian groups [3], expressing non-eosinophilia in nasal polyps with Th1 dominant immunity and prominent neutrophilia. A recent study also corroborated that Th cytokine profiles and eosinophilic/neutrophilic patterns show extreme diversity across different areas and races [4].

At the cytokine level, earlier studies investigated specific cytokines and their mechanisms in polypogenesis, reporting increased IL-6 and IL-8 levels in nasal polyps, and serum IL-16 levels were found to be elevated in patients with nasal polyposis [5,6,7]. In addition, IL-9, IL-10, IL-21, IL-22, and IL-25 were also reported to be involved in nasal polypogenesis [8,9,10,11,12]. Increased IL-36 was found to induce chemokine secretion and increase epithelial permeability in CRS [13], while suppressed IL-37 promoted eosinophilic inflammation in patients with CRSwNP [14]. IL-33 was specifically related to non-eosinophilic CRSwNPs via neutrophil recruitment [15]. However, these studies showed a lack of consistency in inflammatory variables, and most of them concluded that cytokines play a central role in the pathogenesis of nasal polyps. Moreover, investigations into specific cytokines, such as IL-4, IL-5, IL-13, and IL-17A [16], and associated mechanisms facilitate research on novel therapeutic targets and new drug development. However, most of the new drugs were used to reduce only eosinophilic nasal polyps by blocking type 2 cytokines.

As more knowledge about immunological biomarker-directed CRS endotypes has been elucidated, it is believed that some CRS subgroups are prone to developing nasal polyps and postoperative recurrence. Since most previous studies were conducted in Western countries, this study aims to explore the specific cytokines involved in the mechanisms of CRS with recurrent nasal polypogenesis in East Asians.

## 2. Materials and Methods

### 2.1. Study Design and Ethical Considerations

This was a case–control study at a tertiary care otorhinolaryngology center. This study was approved by the Far Eastern Memorial Research Ethics Review Committee (IRB No: 106165-F). Informed consent was obtained from all participants.

### 2.2. Participants

We enrolled patients ranging from ages 20 to 65. From September 2018 to October 2019, we recruited 2 groups of patients for the first experiment: recalcitrant CRSwNP (rCRSwNP) and CRSwNP without recurrence (nCRSwNP), and 3 groups for the second experiment: CRSwNP, CRSsNP and chronic hypertrophic rhinitis (CHR). The diagnosis of CRS was made according to the criteria selected by the American Academy of Otolaryngology Adult Sinusitis Guidelines 2015 [17]. rCRSwNP cases were recruited from postoperative populations who had undergone ESS and followed regularly for at least one year with observed nasal polyp regrowth despite repeated local treatment. In contrast, nCRSwNP cases demonstrated normal sinonasal mucosa without nasal polyps after one year of follow-up. All CRSwNP patients in our second experiment were diagnosed with nasal polyposis and had no history of previous surgical intervention.

Exclusion criteria included known autoimmune disease, unilateral rhinosinusitis, sinonasal tumor or malignancy, antrochoanal polyp, allergic fungal sinusitis, and the use of oral or intranasal steroids within 4 weeks prior to enrollment. Patients with a history of montelukast or any type of immunomodulator therapy (e.g., dupilumab) were also excluded.

### 2.3. Samples and Tissue Preparation

For tissue preparation, nasal polyps from both rCRSwNP and CRSwNP patients were collected; for CRSsNP patients, we obtained sinus mucosa from the uncinate process as specimens. For CHR patients who received septomeatoplasty for nasal blockade, one part of the inferior turbinate mucosa was obtained. For nCRSwNP patients, one sample of mucosa was obtained from the maxillary sinus during follow-up sinoscopy. For the rCRSwNP patients, we examined tissue sections with H-E staining and determined the neutrophil or eosinophil count per high power field (HPF) in three HPFs.

### 2.4. Design and Setting

#### 2.4.1. Cytokine Antibody Arrays

We recruited 12 rCRSwNP patients from our outpatient clinic and compared their tissue interleukin levels to those of 12 nCRSwNP patients (Table 1). We designed a kit of 30 cytokine antibody arrays (RayBio^®^ (Norcross, GA, USA) C-Series Custom Cytokine Antibody Array), including most interleukin families within each subject (Figure 1). The data were normalized and quantified using ImageJ software version 1.53t (National Institutes of Health (NIH), Bethesda, MD, USA).

#### 2.4.2. Enzyme-Linked Immunoassay (ELISA) Experiment

To further identify essential immunological factors relating to nasal polypogenesis, we collected 78 patients and divided them into three groups: CRSwNP (20 cases), CRSsNP (23 cases), and CHR (35 cases). Each patient underwent preoperative sinoscopy and computed tomography (CT). Tissue samples were sent for ELISA study, and blood samples were also obtained if agreed upon by subjects for serum analysis. According to our first experiment, IL-8, IL-15, IL-20, and IL-13 were chosen as biomarkers for comparison among the three patient groups.

#### 2.4.3. IL-8 Production in Human PMNs Treated with IL-7, IL-13, IL-15, and IL-20

Neutrophils are one of the primary cell types that produce IL-8. To determine whether IL-8 production was influenced or enhanced by volumes of other cytokines identified, we isolated polymorphonuclear leukocytes (PMNs) from the tissue and blood of 9 of the 12 rCRSwNP patients and treated the PMNs with IL-7, IL-13, IL-15, and IL-20.

#### 2.4.4. Tissue Immunohistochemistry (IHC) Staining for IL-8 and IL-15

Nasal polyps obtained from one rCRSwNP patient were sent for IHC analysis. Paraffin-embedded tissue slides with microarray samples were used for immunohistochemical staining with antibodies against human CXCL8/IL-8 (Proteintech^®^, Rosemont, IL, USA) and IL-15RA (Proteintech^®^, Rosemont, IL, USA).

### 2.5. Main Outcomes Measures

Comparisons of the expression of cytokine levels between rCRSwNP and nCRSwNP cases are expressed as the median (IQR) and were tested for significance using the M-U test. Due to concerns about multiple tests, we divided the significance level according to the number of tests, and a p-level of less than 0.0017 was regarded as significantly different. We tested expression levels of IL-8, IL-13, IL-15, and IL-20 in both the plasma (p) and tissue (t). Expression levels of the three groups (CRSwNP, CRSsNP, and CHR) in the plasma and tissue are shown as the median (IQR) and were tested with the nonparametric Kruskal–Wallis test. Due to the same concern about multiple testing, we divided the significance level according to the number of tests, and a p-level of less than 0.01 was regarded as significantly different. We further performed a post hoc test, and the significance level was 0.05.

## 3. Results

### 3.1. Comparison of Cytokine Levels between rCRSwNP and nCRSwNP

All 12 rCRSwNP cases demonstrated massive neutrophil infiltration, but 2 cases also had eosinophilia (100 eosinophils per HPF) in three HPFs containing the greatest degree of cellular infiltration [14]. According to the results, IL-8, IL-13, IL-15, and IL-20 demonstrated a significant difference between the rCRSwNP and nCRSwNP groups (*p* < 0.0017, Figure 2A).

### 3.2. Surveillance of Key Cytokine Levels in Tissue and Plasma from CRSwNP, CRSsNP, and CHR Patients

We compared the serum and tissue levels of IL-8, IL-15, IL-20, and IL-13 in 78 patients. The results are shown in Table 2. Tissue IL-8 levels were significantly elevated in CRS patients compared to the non-CRS group. On the other hand, IL-13, IL-15, and IL-20 were not significantly different among the three groups. Regarding tissue IL-8 levels, there was a stepwise significant increase observed in CHR patients (44.8 [78.6], median [IQR] pg/mL) compared to CRSsNP patients (158.8 [194.5] pg/mL) that was further elevated in the CRSwNP group ((291.5 [382.5] pg/mL) *p* < 0.05 in post hoc test) (Figure 3).

### 3.3. IL-8 Production in Human PMNs Treated with IL-7, IL-13, IL-15, and IL-20

Because IL-8 was identified as the key polypogenesis-associated biomarker, the other three cytokines, including IL-13, IL-15, and IL-20, were tested for their modulating effects on IL-8 levels. IL-7, which had borderline significance (*p* = 0.002) in the cytokine antibody array experiment, was also tested. The results are shown in Figure 4, with all values of PMN posttreatment levels standardized to PMN pretreatment levels. For the nine samples obtained from rCRSwNP patients, the IL-8-producing ability of PMNs isolated from nasal polyps was enhanced in response to IL-15 treatment (posttreatment IL-8 level: 3.18-fold elevations compared to the pretreatment IL-8 level) (*p* < 0.05) (Figure 4B). For circulating PMNs isolated from the serum, IL-8 was also elevated after IL-15 treatment (posttreatment IL-8 level: 32.09-fold elevations compared to the pretreatment IL-8 level) (*p* < 0.05) (Figure 4A). Nevertheless, baseline absolute IL-8 levels before the IL-15 challenge were significantly higher in polyp tissue samples than in peripheral blood samples from the nine rCRSwNP patients (*p* < 0.05) (Figure 4C).

### 3.4. Tissue Immunohistochemistry (IHC) Staining for IL-8 and IL-15

In a typical case of rCRSwNP, heavy staining targeting IL-8 was detected at the submucosal layer (Figure 5A). IL-15 was simultaneously detected in the same area when IL-15 was labeled by IHC staining (Figure 5B). Thus, IL-8 production with coexisting IL-15 was evident in a case of rCRSwNP, which was further illustrated by immunohistochemistry double staining (Figure 5C).

## 4. Discussion

In this study, we identified a novel pathway associated with recurrent CRS and nasal polyposis. We employed various approaches to pinpoint key biomarkers related to recurrent CRSwNPs. IL-15 may stimulate PMNs, which secrete IL-8 in the sinus mucosa, predisposing to the recurrence of CRS. This is the first study to demonstrate the IL-15/IL-8 axis in the pathophysiology of CRS and its possible mechanisms in the development of nasal polyps in East Asian patients. In our CRSwNP population that mainly demonstrates neutrophilic NPs, IL-15 may promote IL-8 secretion by PMNs, which further activates downstream inflammatory pathways in the sinus mucosa. The specific targeting of this pathway may represent a new therapeutic strategy for selected CRS patients as an immunomodulatory option.

Recent research focusing on immunohistochemical biomarkers for CRS and the introduction of endotype-driven therapeutic strategies in asthma [18] has enhanced descriptive diagnostic schemes and targeted biologic treatments of CRS at the molecular and cellular levels. It is widely recognized that a new era of drug development has entered the sphere of precision medicine that explores specified biomarkers of individuals treated for CRS [19]. Successful treatment depends on appropriate patient selection and the accurate identification of cytokines, which may be more relevant to disease mechanisms.

IL-8 is a key chemoattractant for both neutrophils and eosinophils in CRS [6]. It is a member of the CXC chemokine family and was released by a variety of cells, such as macrophages, neutrophils, lymphocytes, and endothelial and epithelial cells [20]. This potent chemoattractant is regarded as one of the principal cytokines involved in CRS. It has been demonstrated to be increased in the nasal discharge of CRS patients compared to those from patients with nasal allergies or normal controls [20], and elevated IL-8 levels were detected in the tissue from CRS patients versus normal control mucosa [6]. In addition, it is also believed to be a crucial biomarker in neutrophil-dependent inflammation. In our series, 10 out of 12 recalcitrant CRSwNP cases were neutrophil-predominant. This is in agreement with the conclusions of previous studies where neutrophilic NPs were more frequently observed in Asian populations, while eosinophilic NPs constituted more than 50% of CRSwNPs in Western countries [3,21,22]. Moreover, neutrophil infiltration was also found in eosinophilic polyps in our study, indicating that neutrophilia is commonly found in recurrent CRSwNP in Taiwan but can be accompanied by eosinophilia.

A recent multicenter study demonstrated the universal upregulation of IL-8 levels in CRSwNP patients from all six regions across Europe, Asia, and Oceania; however, more than 50% of patients with CRSwNP show a predominant eosinophilic endotype in Europe and Oceania [4]. IL-8, an essential cytokine for the development of both eosinophilic and non-eosinophilic NPs, is well established. In accordance with previous studies, IL-8 was identified as the most significant biomarker associated with CRS and nasal polyposis in our study using both cytokine protein arrays and ELISA (Table 2 and Figure 2A). Because IL-8 is a strong chemotactic factor, it recruits neutrophils to the site of inflammation, and these neutrophils can become sources of IL-8 release.

Compared to IL-8, the role of IL-15 in CRS is not well-defined. IL-15 was initially identified to share similar activities with its structurally homologous cytokine, IL-2 [23,24], as a stimulant of T-cells, natural killer cells, and innate lymphoid cell proliferation, which plays a pivotal role in innate and adaptive immunity [25]. This pleiotropic cytokine is mainly produced by macrophages as well as various non-lymphoid cells [26]. However, it is constitutively expressed by a large number of cell types, including monocytes, macrophages, dendritic cells, fibroblasts, epithelial cells, and skeletal muscle cells [27]. In addition, disruptions in negative control mechanisms regulating IL-15 expression result in increased IL-15 production, which may predispose to excessive autoreactive T-cell survival and abnormal lymphocyte activation, leading to the development of chronic inflammatory or autoimmune diseases [28]. Using cytokine antibody arrays, IL-15 was determined to be one of the most prominent cytokines identified in rCRSwNPs. Although IL-15 shares similar biological effects with IL-2, IL-2 was not identified as a key biomarker in our study. We sought to clarify the modifying effects of IL-15 on IL-8 to enhance polyp formation and found that PMNs isolated from NPs had the capacity to release IL-8. In response to IL-15 treatment, IL-8 was released from PMNs at a significantly higher level (Figure 4). This finding helps explain why an unpredictable recurrence rate was anticipated in a CRSwNP patient following surgery without a comprehensive understanding of their immune profile, including the level of IL-15. “Surgical removal of tissue IL-8 or medical reduction in local IL-8 with macrolides may reduce its “polypogenic” effects on sinus tissue, but the presence of higher levels of IL-15 could be a “short-cut” for IL-8 augmentation.

A report by A.E. El-Shazly demonstrated crosstalk between the human NK cell and eosinophil recruitment through the IL-15/IL-8 axis in the pathophysiology of allergic rhinitis and indicated the ability of IL-15 to upregulate IL-8 secretagogue activity through NK cells [29]. Richard M. Winn et al. also identified the enhancing effect of IL-15 on IL-8 release by PMNs in response to Aspergillus hyphae [30]. Moreover, it was proven that human neutrophils possess functional IL-15 receptors and generate several functional responses upon IL-15 stimulation and, in particular, IL-8 synthesis and release [31].

Interestingly, differences in IL-8 levels were significant only in tissue samples but not in blood samples in the ELISA experiment (Table 2). However, the enhancing effects of IL-8 production after IL-15 treatment were observed in unchallenged PMNs isolated from both peripheral blood and NPs in the cytokine stimulation test (Figure 4). These results might suggest that the IL-15/IL-8 axis functions in circulation but exerts effects locally and promotes neutrophil-dependent inflammation by increasing IL-8 secretion in sinonasal mucosa. IHC is one of the methods that image discrete immunological and biochemical components and label their anatomical distribution in situ. Using this method, IL-8 and IL-15 were simultaneously labeled in areas of neutrophil infiltration, indicating that the IL-8/neutrophil-rich environment and the presence of IL-15 were evident in our cases (Figure 5(C3)). Indeed, while we reinspected the relationship of these two cytokines in our 12 rCRSwNP patients, IL-8 exhibited a strong positive correlation with IL-15 (r = 0.9184, *p* < 0.05) (Figure 2B).

Notably, in the ELISA experiment, where we detected tissue or peripheral IL-15 levels as well as IL-8, IL-13, and IL-20 levels, only tissue IL-8 levels were significantly higher. Tissue IL-15 elevation was not significant in the CRSwNP and CRSsNP groups, suggesting that CRS may involve complex pathways and that IL-8 enhancement is multifactorial. Thus, the IL-15/IL-8 axis may exert its IL-8-enhancing effect specifically on those recurrent cases in the CRSwNP group. The further expansion of case numbers in our CRSwNP and CRSsNP patients and longitudinal evaluation of their IL-15 levels in recurrent and non-recurrent cases are warranted to clarify the roles of the IL-15/IL-8 axis in these cases.

There are several limitations to this study. First, we enrolled a relatively small number of cases in the cytokine antibody array study. Second, we did not follow patients in the ELISA study postoperatively, which might be the reason for tissue IL-15 levels in the CRSwNP group not being significantly higher than those in the CRSsNP group. Third, for the in vivo functional assessment of the IL-15/IL-8 axis in nasal polypogenesis interpreted from the PMN stimulation experiment and IHC staining, there are limitations inherent to in vitro experiments.

## 5. Conclusions

In this study, we identified IL-8, IL-13, IL-15, and IL-20 as targets for investigating the factors leading to recurrent CRSwNPs using cytokine antibody array screening. Following quantitative analysis using ELISA, IL-8 was identified as a potential proinflammatory factor associated with the development of CRSsNP and CRSwNP in a stepwise fashion in East Asian patients. Due to IL-15’s strong modulating effects on IL-8 levels in our subjects, we propose the IL-15/IL-8 axis as an important pathway contributing to IL-8 accumulation within the sinus mucosa. IL-15 may represent one of the possible underlying cytokines that facilitate IL-8 production and could serve as a potential therapeutic target in patients with recurrent CRS, especially in non-eosinophilic nasal polyps (Figure 6).

## Figures and Tables

**Figure 1 biomedicines-12-00980-f001:**
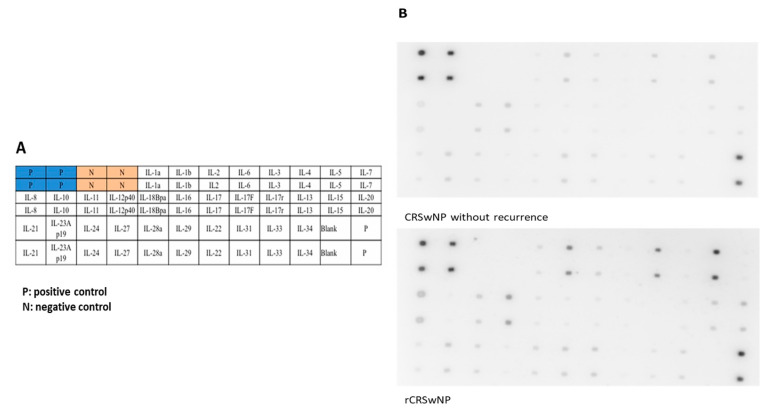
(**A**) The map of cytokine antibodies spotted onto arrays we designed for detecting an individual’s interleukin levels. (**B**) An example demonstrating two kits of cytokine antibody arrays comparing cytokine profiles of nasal polyps from a recurrent CRSwNP patient to those of nasal tissue from an nCRSwNP patient.

**Figure 2 biomedicines-12-00980-f002:**
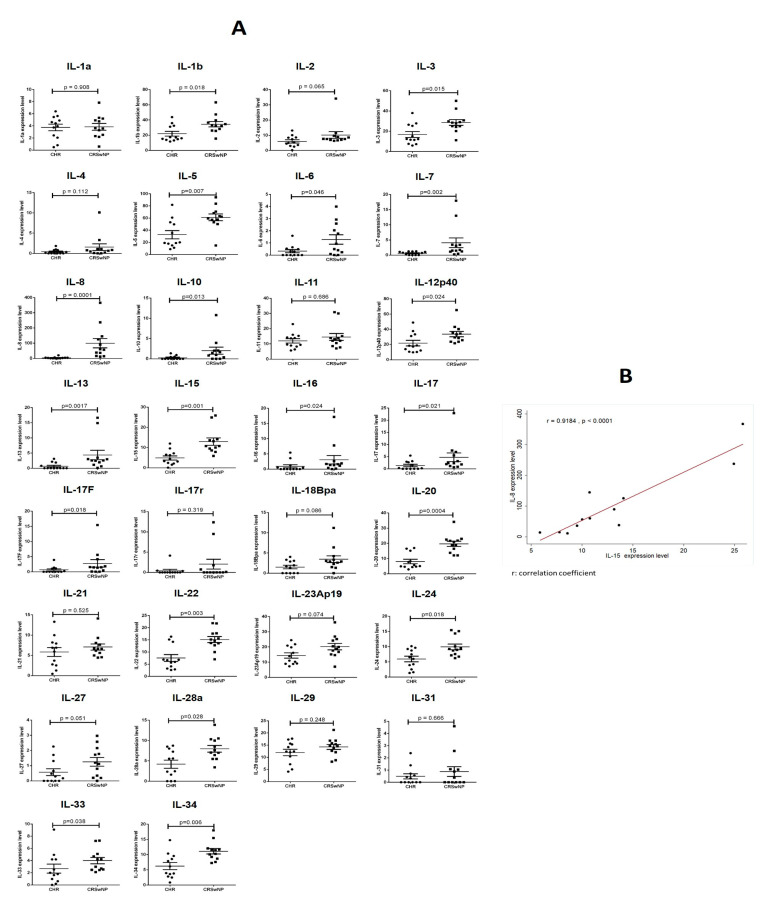
(**A**) Quantitative analysis of cytokine level differences between the two groups: 12 recurrent CRSwNP patients and 12 CRSwNP patients without recurrence were scanned for 30 cytokines. A *p*-value less than 0.0017 was considered to be significantly different. (**B**) The relationship between IL-8 and IL-15 in recurrent CRSwNPs is expressed in a scatter plot, and this correlation is expressed by the correlation coefficient (r). We further tested the correlation coefficient with a linear regression model. The scatter diagram illustrating the relationship between IL-8 and IL-15 revealed a strong positive correlation between IL-8 and IL-15 (r = 0.9184, *p* < 0.001).

**Figure 3 biomedicines-12-00980-f003:**
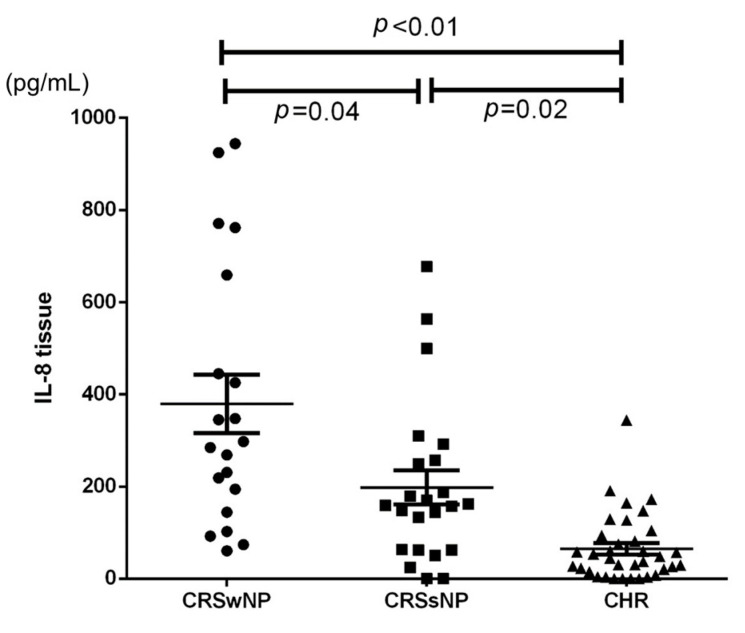
A scatter plot of IL-8 levels revealing elevated levels in the CRS group compared to the CHR group, and a significant increase in IL-8 levels was detected in the CRSwNP group compared to the CRSsNP group.

**Figure 4 biomedicines-12-00980-f004:**
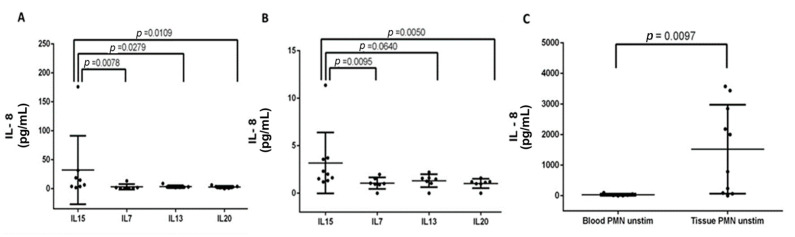
IL-8 production by PMNs is enhanced by IL-15 stimulation. (**A**) Peripheral blood PMNs. (**B**) Tissue PMNs. Data are standardized by comparing one subject’s posttreatment level to its unchallenged state, as shown by the average of subjects tested for different cytokines. (**C**) Absolute IL-8 levels were detected in their unstimulated state, demonstrating that PMNs isolated from polyp tissue produced higher levels of IL-8 than PMNs from peripheral blood.

**Figure 5 biomedicines-12-00980-f005:**
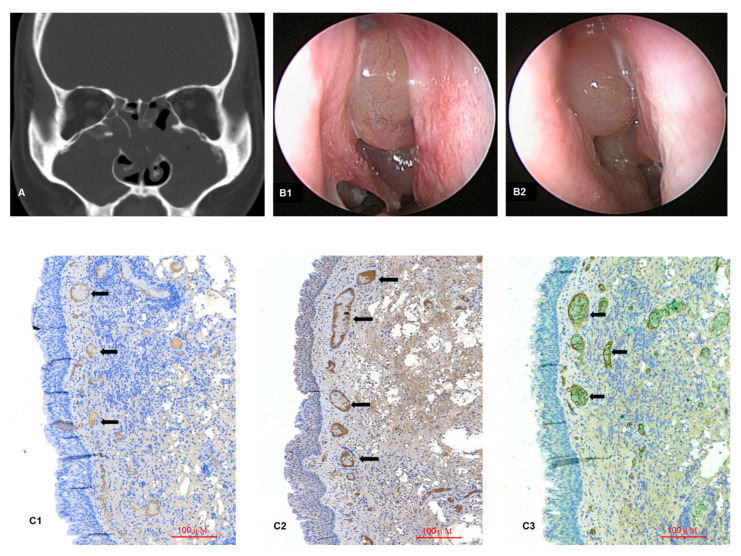
A 30-year-old male patient with prior ESS performed 15 years ago underwent revision surgery for recurrent chronic rhinosinusitis with nasal polyps. (**A**) Preoperative CT scan showing pansinusitis with nasal polyps occupying sinus ostia. (**B**) Recurrent nasal polyposis was noted during follow-up sinoscopy every 6 months (**B1**) until 3 years (**B2**) postoperatively, despite periodic local treatment and medical therapy. (**C**) Tissue immunohistochemistry demonstrated the presence of IL-8 staining, labeled by its antibody with a brown stain (**C1**, arrow); IL-15 staining, labeled by its antibody with a brown stain (**C2**, arrow); and concurrent IL-8 (brown) and IL-15 (green) expressions were found in the same areas with abundant neutrophilic infiltration (**C3**, arrow).

**Figure 6 biomedicines-12-00980-f006:**
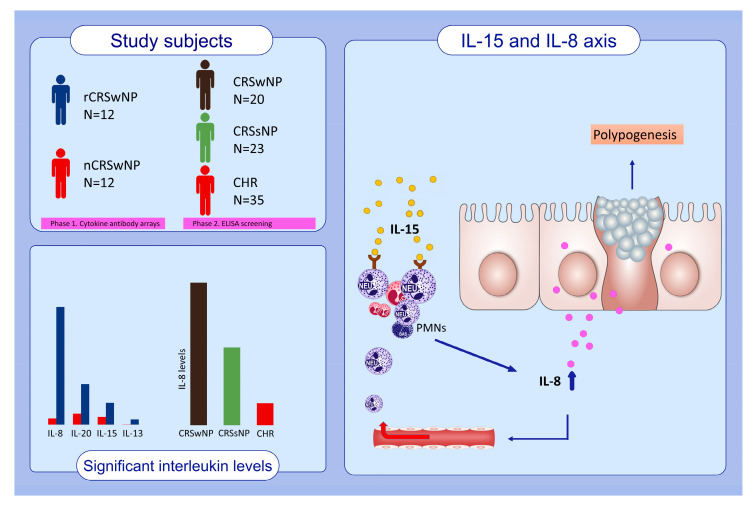
Graphic summary of the current study. Tissue levels of IL-8, IL-20, IL-15, and IL-13 are significantly elevated in rCRSwNP patients. IL-8 emerges as the determinant cytokine for CRS severity, and its secretion from PMNs is significantly enhanced by IL-15. Acting as a potent chemoattractant, IL-8 recruits increasing numbers of PMNs to the sinonasal tissues. Thus, the IL-15/IL-8 axis, mediated via PMNs, provides a potential pathway for regional IL-8 augmentation and promotes polypogenesis.

**Table 1 biomedicines-12-00980-t001:** Demographic data and characteristics of patients scanned with cytokine antibody arrays. Data are expressed as the median (IQR) or number (%).

	rCRSwNP* *n* = 12	nCRSwNP^$^ *n* = 12	*p*-Values ^ξ^
Age (years)	39.5 (20)	29.5 (14)	0.3252
Gender (F:M)	2 (16.7%):10 (83.3%)	1 (8.3%):11 (91.7%)	0.754
IgE (IU/mL)	394 (304)	187 (304)	0.1213
WBC count (/μL)	7340 (2490)	7105 (3990)	0.9540
Eosinophils (%)	2.8 (1.5)	1.2 (2.4)	0.3184
CT score	20.58 (3.34)	0	<0.01

rCRSwNP*: recalcitrant chronic rhinosinusitis with nasal polyp; nCRSwNP^$^: non-recurrent chronic rhinosinusitis with nasal polyp; ^ξ^ Mann–Whitney test.

**Table 2 biomedicines-12-00980-t002:** Plasma and tissue cytokine levels detected by ELISA in three groups, CRSwNP, CRSsNP, and CHR, in order of their significance determined by the cytokine protein arrays experiment, detecting IL-8, IL-13, IL-15, and IL-20; these data are expressed as the median (IQR).

Pg/mL	CRSwNP *n* = 11 (p) *n* = 20 (t)	CRSsNP *n* = 14 (p) *n* = 23 (t)	CHR *n* = 15 (p) *n* = 35 (t)	*p*-Values *
IL-8 p	28.9 (14.2)	37.5 (53.9)	17.0 (28.5)	0.1984
IL-8 t	291.5 (382.5)	158.8 (194.5)	44.8 (78.6)	<0.001 *
IL-13 p	505.5 (34.8)	477.9 (16.3)	483.3 (28.0)	0.0591
IL-13 t	305.1 (294.8)	261.8 (300.1)	513.2 (273.3)	0.7018
IL-15 p	44.4 (4.3)	43.4 (6.6)	39.6 (9.4)	0.3934
IL-15 t	63.8 (27.1)	55.1 (20.5)	52.5 (25.0)	0.3292
IL-20 p	- ^$^	0 (0.6)	0 (0.1)	0.3609
IL-20 t	1.0 (2.4)	1.9 (5)	1.9 (4.1)	0.2094

p: plasma; t: tissue; * Nonparametric Kruskal–Wallis test; ^$^ very low; CRSwNP: chronic rhinosinusitis with nasal polyp; CRSsNP: chronic rhinosinusitis without nasal polyp; CHR: chronic hypertrophic rhinitis.

## Data Availability

Data are contained within the article.

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
