# Peer review of "The Interleukin-15 and Interleukin-8 Axis as a Novel Mechanism for Recurrent Chronic Rhinosinusitis with Nasal Polyps"

_biomedicines, 2024, doi:10.3390/biomedicines12050980_

Round 1

Reviewer 1 Report

Comments and Suggestions for Authors

This manuscript is well written according to author's intensive research data on mucosal linings of chronic rhinitis patient.  I reccommend  this paper can be accepted for publication in MDPI journal, but authors should need much more discussion as regard IL-15.  This pleiotropic cytokine reveals various types of function to mucosal site of inflammation, such as allergic rhinitis, bronchial asthma and so on.  For example, is IL-15 produced from what kinds of cells, Just NK cells or epithelial cells, and why IL-15 production is upregulated in mucosal linings of rCRSwNP. Furthermore, what is the result of IL-15 in mucosal linings of CRSsNP, not persistent ? Otherwise, IL-8/IL-15 axis, which authors proposed firstly, is not accepted from their results. Authors should revise this paper as much as they can from this point of view. However, nonetheless, I highly evaluate this work as nice piece of paper. 

Reviewer 2 Report

Comments and Suggestions for Authors

The authors have conducted a two phase study in patients with nasal polyposis and included different subgroups and identified IL-8 as an important pathway.

It is a well conducted study and a few clarifications will help to make it stronger

figure 2 is not clear

In the second phase, how many of these patients had recurrent nasal polyposis or were all cases had nasal polyps for the first time

In the entire group, how many had been treated with intranasal steroids, montelukast and biologics specially dupilumab

Add a few sentences on the strength of the study

Write conclusions for phase I and phase II of the study separately. They are not exactly the same population and the conclusion therefore will be different 

Comments on the Quality of English Language

minor editing needed

Round 2

Reviewer 1 Report

Comments and Suggestions for Authors

I think no more revision is needed to be accepted, even though I am not fully satisfied with author's revised manuscript.